# Advances in Recapitulating Alzheimer’s Disease Phenotypes Using Human Induced Pluripotent Stem Cell-Based In Vitro Models

**DOI:** 10.3390/brainsci12050552

**Published:** 2022-04-26

**Authors:** Md Fayad Hasan, Eugenia Trushina

**Affiliations:** 1Department of Neurology, Mayo Clinic, Rochester, MN 55905, USA; hasan.mdfayad@mayo.edu; 2Department of Molecular Pharmacology and Experimental Therapeutics, Mayo Clinic, Rochester, MN 55905, USA

**Keywords:** Alzheimer’s disease, biofabrication, disease modeling, human induced pluripotent stem cell (HiPSC), microfluidics, organoid, spheroid, stem cells, 3D culture

## Abstract

Alzheimer’s disease (AD) is an incurable neurodegenerative disorder and the leading cause of death among older individuals. Available treatment strategies only temporarily mitigate symptoms without modifying disease progression. Recent studies revealed the multifaceted neurobiology of AD and shifted the target of drug development. Established animal models of AD are mostly tailored to yield a subset of disease phenotypes, which do not recapitulate the complexity of sporadic late-onset AD, the most common form of the disease. The use of human induced pluripotent stem cells (HiPSCs) offers unique opportunities to fill these gaps. Emerging technology allows the development of disease models that recapitulate a brain-like microenvironment using patient-derived cells. These models retain the individual’s unraveled genetic background, yielding clinically relevant disease phenotypes and enabling cost-effective, high-throughput studies for drug discovery. Here, we review the development of various HiPSC-based models to study AD mechanisms and their application in drug discovery.

## 1. Introduction

Alzheimer’s disease (AD) is a multifactorial disorder without a cure. In 2021, the yearly cost of AD was anticipated to reach $355 billion, with a steep rising curve [1]. An estimated 6.2 million adults over 65 years of age are now living with this devastating disease, with a projected 2-fold increase in patient number by 2050 [1]. These numbers ardently necessitate the development of therapeutics that could halt disease progression. A commensurate amount of effort is being employed in this field. Novel underlying AD mechanisms have been unearthed, forcing researchers to question the role of amyloid-β (Aβ) and phosphorylated tau (pTau) protein as major culprits in AD pathogenesis [2,3,4,5]. Controversial [6,7] results of clinical trials on reducing Aβ [8,9] or pTau levels [3,4,5] refocused AD drug development. Increasing importance is now being given to inflammation, synaptic dysfunction, altered metabolism, neurogenesis, and epigenetics [10]. In fact, 126 clinical trials in 2021 targeted mechanisms other than Aβ and pTau [11]. Multiple new and repurposed compounds focused on early disease mechanisms are currently in preclinical trials. The availability of models that accurately recapitulate disease phenotypes in a patient-specific manner could play a critical role in successful translation of experimental approaches to human trials. Historically, animal models were extensively used in AD research to test preclinical candidate compounds. AD phenotypes in these models were mainly recapitulated by the overexpression or ablation of genes of interest [12]. Particularly, transgenic mouse models with the overexpression of human mutant amyloid precursor protein (APP) or presenilin 1 (PSEN1) protein, both involved in Aβ production [13], were used the most [12]. However, these genetic mutations are found only in rare familial variants of AD (FAD). The most prevalent form of AD, sporadic late-onset AD (LOAD) [14], is associated with multiple risk factors, including age, female sex, and the presence of the apolipoprotein E4 (*apoE4*) allele, which makes modeling of LOAD in mice very difficult. While new National Institutes of Health (NIH) programs (e.g., MODEL-AD [15]) were initiated to develop animal models of LOAD, there is a necessity for complementary models that recapitulate LOAD phenotypes for screening of new compounds at an early stage of the drug development pipeline.

The recent emergence of stem cell technology provides an outstanding opportunity to address this need. The 2012 Nobel Prize–winning work of Shinya Yamanaka, MD, PhD, showed that human fibroblasts can be reprogrammed to become pluripotent [16]. Though embryonic stem cells have been studied for 2 decades [17], the use of human induced pluripotent stem cells (HiPSCs) not only removes source restrictions but also mitigates ethical concerns [18]. While modeling AD, fibroblasts can be easily obtained from patients by minimally invasive superficial skin biopsy. HiPSCs are generated from these fibroblasts by overexpressing pluripotency factors using cell reprogramming [16]. HiPSCs are then directed by appropriate growth factors [19,20], small molecule cocktail [21,22], or lentiviral overexpression of transcription factors [23,24] to differentiate into various types of neurons, astrocytes, or microglia [25]. Alternately, just like in animal models, known genes involved in FAD could be overexpressed in these HiPSCs using clustered regularly interspaced short palindromic repeats (CRISPR) and CRISPR associated protein 9 (CRISPR/CAS9). Cells obtained from differentiating these modified HiPSC lines have FAD-associated mutations. These cells with AD traits, either directly from the patient or by forced expression, are then cultured in different setups that span from simple 2-dimensional (2D) cultures to organoid and complex microfluidic device–based cocultures. Here, we review multiple techniques developed to date to model AD in vitro, their advantages and disadvantages, and future directions.

## 2. HiPSC-Based Models of AD

The power of HiPSCs lies in their ability to retain unique genetic traits of an individual. This technology also can capture the acquired epigenetic modifications associated with environmental exposure. These advantages make HiPSC technology exceptionally suited to study the mechanisms of LOAD and to establish patient-specific therapeutic development platforms. Fibroblasts collected from individuals using skin biopsy are converted into HiPSCs by an overexpression of 4 transcription factors (octamer-binding transcription factor 4, Krüppel-like factor 4, SRY-box transcription factor 2, and c-myelocytomatosis oncogene product) [16]. HiPSCs can be almost indefinitely expanded and passaged to inexpensively obtain large numbers of cells. Multiple methods have been developed to differentiate mature and functional neurons [26], astrocytes [27], and microglia [28] from these HiPSCs. Initially, these differentiation protocols were formulated by mimicking embryonic development. Dual SMAD (an acronym from the fusion of the *Caenorhabditis elegans* and *Drosophila* genes, *Sma* and *Mad* [mothers against decapentaplegic]) inhibition is one of the most well-established protocols. SMAD is a group of proteins that are crucial for cell development and growth. By systematically inhibiting unwanted germ layer (mesendoderm and trophectoderm) growth using proteins [20] and/or small molecule cocktail [29], the HiPSCs are forced to assume neuroectodermal fate. Tailored growth factor combination is then used to differentiate these precursor cells into selective subtypes of neurons (e.g., glutamatergic or dopaminergic) and glial cells. These methods are especially useful where brain development is under scrutiny.

Brain-like organoids were also developed using this technique [30]. Though constantly improved by ongoing studies, this method does not yield a pure population of cell types and requires the use of advanced purification systems, such as fluorescence-activated cell sorting (FACS). This makes analysis of specific cell type attributes difficult. Multiple transcription factor or micro-RNA–based methods have also been developed to yield a pure population of desired brain cells (e.g., cortical neurons [23] or astrocytes [24]) from HiPSCs. Tetracycline-inducible expression of transcription factors can be used to rapidly stimulate neuronal [23] or astrocytic [24] differentiation in a controlled manner. Antibiotic purification using puromycin, hygromycin, or blasticidin yields a pure population of desired cell types. Striatal medium spiny neurons have been differentiated from fibroblasts using a similar method [31]. These striatal cells, when differentiated from cells from patients with AD, are reported to retain host genetic traits and express AD-like phenotypes. Alternately, CRISPR/CAS9 [30] can be used to introduce AD-related mutations in HiPSCs. Cells differentiated from these modified HiPSCs could also model AD in vitro.

These cells with AD-related traits are then cultured separately or together in different setups, ranging from simple 2D to complex 3-dimensional (3D) cultures and brain organoids. Enhanced brain mimicry and freedom over mechanical cues in these cultures allow a closer look into mechanisms of AD in systems with human origin. Instead of generalizing an AD phenotype for all patients, these approaches aim to understand AD in a patient-specific manner, contributing to individualized medicine approaches. In general, these in vitro techniques to study AD could be divided in 3 categories: (1) HiPSC-derived 2D models, where monolayer cells are cultured on a flat surface; (2) HiPSC-derived organoid models, where cell (HiPSC) aggregates are differentiated in a controlled environment to yield 3D constructs comprising multiple cell types; and (3) engineered 3D models, where differentiated brain cells are engineered by mechanical cues to create 3D constructs with predefined attributes. Herein, we describe the advantages and limitations of these models.

### 2.1. HiPSC-Derived 2D AD Models

#### 2.1.1. Neuron-Focused 2D AD Models

AD is marked by initial episodic memory loss and gradual severe cognitive decline [32]. Historically, AD progression and cognitive decline have been best correlated with neurodegeneration and neuronal dysfunction [33,34]. This observation has set the initial focus of AD models towards neurons. After the initial establishment of HiPSC technology in 2007, most of the HiPSC-based AD models adopted a minimalist 2D approach where patient-derived stem cells are differentiated to yield 2D monolayer neuronal cultures. When derived from patients with FAD, these cultures present consistent AD-related phenotypes, including early endosomal dysfunction [35], elevated levels of Aβ and pTau [35], increased Aβ_42_/Aβ_40_ ratio in culture medium [36,37], and activation of glycogen synthase kinase-3β (GSK3β) [35]. However, these phenotypes have only been shown to emerge in cells derived from a subset of patients with LOAD [35,36]. Since the development of LOAD can be influenced by epigenetic factors [38], the conversion from the patient’s fibroblasts to neurons may not retain these traits. Another possibility includes failure to recapitulate delayed onset of LOAD phenotypes in immature (21–180 days) cultures. This also can be true for FAD cultures. In one study, endoplasmic reticulum and oxidative stress [36] were not observed in 21-day-old FAD cultures but were found in 180-day-old cultures. Regarding the generation of Aβ, studies show that the β-secretase inhibitor, β-site APP cleaving enzyme 1 (BACE1), can reverse adverse Aβ pathology [36,37,39]. Interestingly, endosomal dysfunction in neuronal cultures with FAD mutations are shown to be mediated by this amyloidogenic processing of APP (β-C-terminal fragment [β-CTF]) but not Aβ itself [40]. These Aβ pathologies are prominent in FAD-derived but inconsistent in LOAD-derived cultures. However, the identification of the AD risk factor *apoE4* allele has been shown to improve the consistency of the outcomes [41] contributing to the reconsideration of a causative role of Aβ in AD [42].

Accumulation of pTau, another hallmark of AD, was consistently found in neuronal cultures derived from patients with FAD and patients with LOAD [35,43]. Neurons differentiated from HiPSCs of patients with *apoE4* allele (*apoE4^+^*) and LOAD using lentiviral overexpression of transcription factor neurogenin-2 [23] demonstrated increased levels of pTau and activation of extracellular signal-regulated kinases 1 and 2 (ERK1/2). Interestingly, this tauopathy was ameliorated by gene substitution from *apoE4* to *apoE3*. These pure neuronal cultures were more sensitive to Aβ oligomers or hydrogen peroxide (H_2_O_2_) -induced oxidative stress compared to their healthy patient–derived counterparts [23]. However, similar to previous findings, the Aβ_42_/Aβ_40_ ratio was elevated only in cultures derived from patients with FAD but not LOAD. While some studies claim there is a correlation between Aβ and pTau [35], pTau has been independently correlated with cholesteryl esters, the storage product of cholesterol excess [44]. This study also identified 2 different pathways through which cholesteryl esters affect Aβ and pTau (cholesterol binding site in APP and proteosome, respectively). This model was instrumental in identifying compounds that could alleviate tauopathy by targeting cholesterol metabolism pathways. Furthermore, docosahexaenoic acid, an omega-3 fatty acid, was shown to be effective in reducing Aβ oligomer-induced oxidative stress in these cultures [36]. These results support the importance of abnormal lipid homeostasis in AD pathogenesis and demonstrate the utility of 2D neuronal models for therapeutic development.

Endocytosis and transcytosis of low-density lipoproteins were impaired in neurons differentiated from healthy HiPSCs with mutant *PSEN1* introduced using CRISPR/CAS9 [45]. Guanosine triphosphate hydrolase Ras-related protein encoding gene *Rab11*, an important regulator of vesicle recycling and transcytosis [46], was found to be downregulated in axons and upregulated in the soma of these neurons [45]. Ras-related protein 5a encoding gene *Rab-5*, an early endosome enlargement marker, was also found in abundance in neurons from *APP* and *PSEN1* knock-in HiPSCs [40]. These studies conform to previous works [40,45,46] demonstrating that endosomal dysfunction in neurons is mediated by amyloidogenic processing of APP (β-CTF fragments) but not Aβ itself [45] and application of BACE1 inhibitor improves endosomal abnormalities. These results emphasize the role of nonamyloid pathways in AD pathophysiology, the involvement of endosomal and axonal trafficking abnormalities in particular.

Mitochondria were recently stated as a viable target for therapeutic development for AD [47,48,49]. Mitochondrial dysfunction [48] is found in AD brain prior to the emergence of Aβ or pTau pathology [50,51]. Neuronal cultures derived from patients with LOAD recapitulated this trait [52]. Aberrant reactive oxygen species production was observed in LOAD cultures, which correlated with DNA damage but not Aβ and pTau accumulation. These neurons exhibited altered oxidative phosphorylation (OXPHOS) and reduced resistance to H_2_O_2_ injury [53]. Mitophagy, a process involved in the removal of damaged mitochondria, was also shown to be affected in these cultures, leading to the accumulation of defective mitochondria [54]. An additional study showed an increase in dysfunctional lysosomes, defects in OXPHOS, and aberrant mitophagy in neurons derived from patients with FAD with *PSEN1* A246E mutation [55]. These results corroborate previous findings and establish mitochondria as a potential therapeutic target in AD [56,57].

In early stages of AD, human neurons exhibit hyperexcitability [58]. The recapitulation of this phenotype in vitro requires a presence of functional synaptic networks. Neurons differentiated from HiPSCs with AD-related mutations in *APP* or *PSEN1* (introduced using CRISPR/CAS9) yielded this phenotype [59]. These neurons exhibited increased frequency of spontaneous action potentials, evoked activity, altered action potential shape, and shorter neuritic processes. Supporting hyperexcitability in AD, ɣ-aminobutyric acid–mediated (GABAergic) neuron-specific degeneration was also found in cortical [59] and forebrain interneurons [60,61] differentiated from *apoE4^+^* LOAD HiPSCs. FAD HiPSC-derived GABAergic neurons also showed altered functionality, enhanced levels of pTau, activation of stress response pathways, and upregulation of neurodegenerative pathways [62]. Possible deficiencies in chloride exporter KCC2 (SLC12A5) in these neurons hindered the achievement of chloride ion reversal potential, leading to deficient neuronal inhibition. Both gene substitution from *apoE4* to *apoE3* and small molecule structure correctors were effective in ameliorating GABAergic neuron-specific degeneration [61]. Cholinergic neurons derived from patients with LOAD also showed increased susceptibility to glutamate-mediated cell death [63], increased Aβ_42_/Aβ_40_ ratio [64], and altered calcium ion (Ca^2+^) flux [64].

Taken together, this body of evidence establishes 2D monolayer cultures as an excellent tool to assess a single cell-specific neuronal AD phenotype. These cultures are relatively simple and produce network level insights relevant to neuronal activity. However, the absence of glial cells, essential for synaptic maturation [65] and inflammation, can be marked as one of the major limitations of these systems.

#### 2.1.2. Astrocyte-Focused 2D AD Models

Though neuron-focused models can capture multiple aspects of AD, other cell types, including astrocytes [66,67] and microglia [68], also actively contribute to disease development and progression. Microglia and astrocytes are key players in neuroinflammation, which is a contributing factor to AD pathogenesis [69,70]. In the systems described in Section 2.1.1, astrocytes derived from patients with LOAD or FAD were not used. Evidently, these approaches overlooked the important astrocyte-specific disease phenotypes. This gap is primarily due to the difficulty in differentiating astrocytes using conventional methods where neural progenitor cells (NPCs) are obtained from HiPSCs by dual SMAD inhibition [20]. This inhibition directs the NPCs towards astrocytic fate. NPCs are then expanded until gliogenesis [71,72]. This process is time consuming (requiring more than 6 months [73]) and yields a heterogenous mixture of different cell types. Additional FACS is often required to isolate cells of interest [19].

Regardless of the aforementioned difficulties, the importance of astrocytes in the pathogenesis of AD was demonstrated in the HiPSC-derived cocultures of patients with early-onset FAD where astrocytes exhibited altered metabolism, increased oxidative stress, and disturbed Ca^2+^ signaling in the endoplasmic reticulum [74,75]. In line with AD-related defects in neuronal lipid metabolism [76], 2D astrocyte cultures from patients with FAD showed impaired fatty acid oxidation [77], altered morphology [78], and nonstimulated release of soluble inflammatory mediators [78]. Fatty acid oxidation impairment in 2D astrocyte cultures was corrected by synthetic peroxisome proliferator activated receptor-β and -δ (PPARβ/δ) agonist GW0742 [77]. PPARβ/δ is a key player in brain energy homeostasis and metabolism. Further studies showed that, in astrocyte-neuron cocultures, these astrocytes provide reduced support towards neuronal survival and synaptogenesis, exhibit reduced glucose uptake [79,80], and oversupply cholesterol to neurons causing neuronal lipid raft expansion and Aβ_42_ production [81]. Altered bioenergetic metabolites and transcriptomes were also reported in these astrocytes [80].

Taken together, astrocyte-focused models of AD establish astrocytes as key players in the pathogenesis of AD, which supports multiple neuronal AD phenotypes, including a decreased resistance to H_2_O_2_ injury and defects in lipid metabolism. An inflammatory AD phenotype, absent in neuron-only models, was recapitulated in the presence of astrocytes. However, as mentioned earlier, these models were mainly based on dual SMAD inhibition of HiPSC-derived NPCs and time-consuming gliogenesis. Inherent heterogeneity of cell types makes these systems unreliable and difficult to control.

#### 2.1.3. Microglia-Focused 2D AD Models

The involvement of microglia in AD neurobiology has been well established [82]. However, the elusive embryonic origin of microglia has impeded the development of reliable protocols for their differentiation from HiPSCs. Due to this, past studies were limited to rodent, hard-to-obtain ex vivo human microglia [83], or immortalized cell lines, yielding cultures with substantially different characteristics [84]. Recently, primitive myeloid progenitors were identified as the origin of microglia [85], differentiating these cells from other tissue-resident macrophages originating from yolk sac–derived erythromyeloid progenitors [86]. This ontogenetic advancement has recently led to the development of the first microglia differentiation protocol [87]. 2D cocultures from *apoE4^+^* neuron [23], astrocyte [88], and microglia [87], derived from patients with LOAD [23], were developed using this method [89]. Concurring with neuron-only models, neurons in this triculture setup exhibited an increased number of synapses and elevated neuronal Aβ_42_ secretion. Astrocytes exhibited impaired Aβ uptake and cholesterol accumulation, which was shown to increase astrocytic cholesterol secretion, leading to neuronal lipid raft expansion and Aβ_42_ production [81]. Importantly, microglia exhibited altered morphology, impaired Aβ phagocytosis [89], and mutual astrocyte-microglia activation [90].

Recent development of small molecule–guided [91] microglial differentiation has further accelerated studies with cocultures [92]. Microglia differentiated from HiPSCs derived from patients with LOAD and FAD using small molecules exhibited *apoE4* genotype–induced aggravated microglial inflammatory response, decreased microglial metabolism, phagocytosis, and migration. Counterintuitively, microglia generated from HiPSCs with FAD-related mutations in *APP* and *PSEN1* showed only a slight decrease in proinflammatory cytokine release and increase in chemokines, indicating a senescent-like state [92], while microglia derived from HiPSCs from patients with LOAD with *apoE4* allele assumed a more proinflammatory state [92]. These results indicate the possibility of mechanistic differences between LOAD and FAD [92].

Advancement in stem cell differentiation techniques and automated culturing setups have further allowed the development of a high-throughput and automated HiPSC-derived cell culturing platform [93]. Using this platform, highly replicable 2D neuron-astrocyte-microglia tricultures were created in 384-well plates [93]. Application of soluble Aβ_42_ in these tricultures resulted in Aβ plaque formation with surrounding dystrophic neurites [93]. The *apoE4/3* microglia were found to internalize, exocytose, and package soluble Aβ_42_ as plaque structures [93]. Hence, microglia not only surrounded these plaques, but participated in plaque formation. Multiple other AD-related phenotypes, such as synapse loss, dendritic retraction, axonal fragmentation, pTau accumulation, and neuronal cell death, were also reliably recapitulated in these cocultures. The highly automated and precise nature of this setup made it a convenient tool for high-throughput compound screening. Seventy neuroprotective small molecule compounds (4 concentrations in at least 2 independent cultures for each compound) were screened in this platform, using the improvement in dendritic (microtubule-associated protein 2 positive) area, axonal (βIII-tubulin positive) area, number of synapses (synapsin 1/2 positive puncta count), and cell survival (cut like homeobox 2 positive cell count) as the therapeutic index. Nine hits were identified in this screening, including inhibitors of well-known active kinases in AD, such as DLKi27 [94], indirubin-3′-monoxime [95] (GSK3β and CDK5 inhibitor), AZD0530 [96] (Fyn inhibitor), and demeclocycline HCl [97] (calpain inhibitor), along with luteolin [98,99] and curcumin [99] and its derivative J147 [100]. Involvement of the DLK-JNK-cJun pathway in AD was further confirmed by validating the neuroprotection of VX-680 [101] (a different DLK inhibitor), GNE-495 [102] (MAP4K4 inhibitor upstream of DLK45), PF06260933 (a different MAP4K4 inhibitor), and JNK-IN-8 (JNK1/2/3 inhibitor) [93]. Neuroprotective effects of anti-Aβ antibody were also validated using this setup. These results show the potential and capacity of a full-fledged, high-throughput system that uses HiPSCs and automation technologies.

Table 1 summarizes 2D AD models that recapitulate multiple AD phenotypes (Figure 1) and shed light on previously unraveled mechanisms. The addition of astrocytes and microglia considerably enhance AD modeling in these cultures. However, 2D in vitro models lack brain-like complexity, microenvironment, and extracellular matrix (ECM) [103]. Cells in this flat setting also exhibit immature metabolism and neuronal activity [104,105]. It was further shown that neuronal cytoskeleton redistribution in response to Aβ oligomers cannot be recapitulated in 2D neuronal cultures [106]. To mimic human brain with enhanced accuracy, organoid, spheroid, and engineered 3D culturing techniques have recently emerged.

### 2.2. HiPSC-Derived Organoid Models of AD

The addition of the extra dimension to conventional 2D cultures started with spheroid cultures. Spheroid cultures are cell aggregates embedded in an artificial ECM. This cellular aggregation can be induced by a low attachment substrate [109] or confinement [110,111]. The primary hurdle of this technology is core necrosis. Since organoids are not vascularized [112], nutrients cannot reach deep inside cell clusters. Cellular waste also cannot be efficiently removed from the deep layers of cell aggregates. As a result, there is upward cellular waste and downward nutrient gradient from the spheroids’ outer layer to the center, resulting in a necrotic core which limits culture size. Advanced biocompatible ECM or scaffolds partially allow nutrients to enter deep inside the cultures and help to overcome core necrosis. Cells are generally embedded in these scaffolds and then differentiated. Recently emerged bioreactor [113,114] technology employs highly controlled rotating walls, creating optimum medium flow that negates gravitational pull to create microgravity or weightlessness. Scaffold-embedded cells are then differentiated under these conditions, which promote aeration, uniform cellular aggregation, and enhanced nutrient diffusion. In the case of HiPSC-derived spheroids, cells in these aggregates are differentiated into their fates determined by supplied growth factors. Interestingly, during the differentiation process, the gradually changing microenvironment can influence cells to assume brain-like laminar positions. These spheroids are termed brain organoids and are defined as 3D structures derived from either pluripotent stem cells (embryonic stem cells or HiPSCs) or neonatal or adult stem/progenitor cells, in which cells spontaneously self-organize into properly differentiated functional cell types and recapitulate at least some function of the organ [115].

Existing 3D HiPSC-based AD models use ECM to hold cells in their 3D locations. NPCs or HiPSCs embedded in Matrigel (Corning Life Sciences, Bedford, MA, USA) can be differentiated to yield homogeneous cortical neuronal spheroid cultures [116]. Neuron-specific adeno-associated viral overexpression of P301L (tau mutation most frequently observed in patients with frontotemporal dementia and parkinsonism [117]) and application of recombinant human tau (K18) yielded tauopathy in a Matrigel-embedded setup [116] (Figure 2**,** top row). In the same system, immortalized human NPCs (ReNcell VM, ReNeuron Group plc) with FAD-related mutations in *APP* and *PSEN1* yielded homogeneous neuronal cultures with extracellular deposition of Aβ plaques, Aβ_42_/Aβ_40_ ratio–correlated tauopathy, and cell death [118] (Table 2). Interestingly, Aβ plaques were found in 2D neuron-astrocyte-microglia triculture only after application of synthetic Aβ oligomers [93], while in FAD 3D cultures they appeared spontaneously. Compared to 2D cultures, induced neurons also showed enhanced maturation in this 3D setup [118,119]. However, these spheroid cultures did not show any brain-like cellular organization or layers.

Further progress in brain organoid technology has produced methods to mimic corticogenesis (the developmental process of the cerebral cortex) in a dish and create neocortex-like 3D cultures from HiPSCs [120]. In these systems, HiPSCs are first self-aggregated in low-adherent, concave (V- or U-bottom) wells to generate embryoid bodies. These embryoid bodies are maintained long term in the media with rho kinase inhibitor, low concentration Matrigel, and other growth factors [120]. Next, the transforming growth factor β and wingless/Int-1 signaling pathways are chemically blocked to induce neuroepithelium differentiation on a low-adherent substrate (Figure 2, middle row). This method yielded spherical structures (spheroids) that exhibited brain-like cortical regions, including progenitor zones and ventricular zones (Figure 2, middle row, right), termed cerebral organoids (COs). HiPSC-derived COs [121,122,123,124] from patients with FAD and LOAD [89] were characterized with Aβ plaques, elevated pTau, mislocalized mutant tau protein, and dysfunctional axons. These phenotypes were cross-validated in 2D cultures, COs, cerebral spinal fluid, and ex vivo brain tissue from the same patient with an FAD-linked APP mutation [125]. Mitochondrial axonal transport [126] was impaired in COs generated from patients with familial frontotemporal dementia with R406W (known cause of frontotemporal dementia with parkinsonism [127]) tau mutation. This phenotype was absent in previous systems. Moreover, microtubule stabilization rescued axonal transport deficiency, indicating a contribution of mitochondrial motility to the AD mechanism. Furthermore, the process of neuronal differentiation was impaired by the repressor element 1-silencing transcription factor (*REST*) gene in COs derived from patients with LOAD [128]. This impairment was unaffected by Aβ production, BACE1, and γ-secretase inhibitors [128]. These observations shed light on possible developmental defects in AD and reestablishes *REST*, a regulator of aging brain stress response, as a viable marker of AD [129]. Hippocampal organoids were also developed by chemical differentiation of HiPSC spheroids in suspension [130,131]. Similar to COs, these hippocampal organoids showed increased Aβ_42_/Aβ_40_ ratio and decreased levels of synaptic proteins [130,131].

**Figure 2 brainsci-12-00552-f002:**
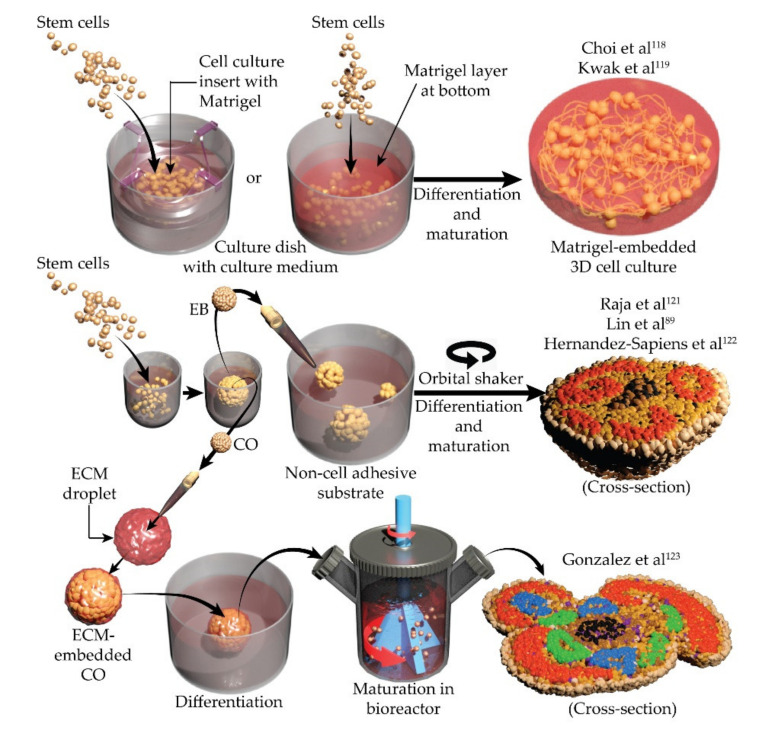
Simplified description of processes used to create 3D cultures for modeling Alzheimer’s disease in vitro [89,118,119,121,123]. CO, cerebral organoid; EB, embryoid body; ECM, extracellular matrix; 3D, 3-dimensional.

The 3D, spheroid, and organoid models of AD have added a new dimension to conventional 2D models. Notably, the hallmark of AD, the formation of Aβ plaques, was readily observed in these models. Organoids also exhibited AD-related defects in axonal trafficking. Both traits cannot be mimicked in 2D cultures without the application of synthetic Aβ peptides. Since AD is a multifactorial disease, efficacious therapeutic strategies would be expected to improve multiple pathways. Spheroids and especially organoids present a wholesome accumulation of almost all known AD phenotypes in a single model. This makes them a valuable tool for drug discovery compared to conventional 2D cultures. While 2D cultures continue to be the most convenient model to assess single cell attributes, organoids are rapidly recognized for their ability to provide input on a systems level. Interestingly, in this 3D system, organoids were able to recapitulate AD phenotypes (Table 2) in months, which takes years to develop in a human brain [125].

Despite their promises, brain organoids face some crucial problems that hinder their use in high-throughput drug screening applications. Since organoids are differentiated from HiPSCs or NPC aggregates without any purification step, cellular composition in these cultures cannot be exactly predefined [132,133]. Probably because of these discrepancies, neural activity also is not well defined in organoids [134]. In addition, culture morphology cannot be designed with these suspension setups [135]. Cellular and morphologic heterogeneity result in substantial culture-to-culture variations [125]. Scaffold-based optimized variants of these cultures, called *mini brain* [136], develop brain-like regions. However, these regions also vary considerably in shape and location among cultures. These problems require more controlled and defined 3D culturing techniques.

**Table 2 brainsci-12-00552-t002:** Spheroid and organoid models of AD.

Observed AD Phenotype	Method	AD Source	Experimental Time Point, Days	Reference
Neurofibrillary tanglelike inclusions	Dual SMAD inhibition and Matrigel (Corning Life Sciences)—embedded 3D maturation	Application of recombinant human tau (K18) to P301L overexpressed neurons differentiated from NPCs	>28	[116]
Extracellular deposition of Aβ, including Aβ plaques Aggregates of pTau in the soma and neurites and filamentous tau	Matrigel-embedded differentiation [137]	Lentiviral overexpression of FAD-related mutations in *APP* and *PSEN1* of ReNcell VM (ReNeuron Group plc)	49–100	[118]
Aβ_42_/Aβ_40_-correlated increase of pTau and cell death	Matrigel-embedded differentiation [138]	Lentiviral overexpression of FAD-related mutations in *APP* and *PSEN1* of ReNcell VM and FACS purification	35–84	[119]
Aβ accumulation and elevated pTau	Matrigel-embedded self-organized differentiation	FAD (*APP* and *PSEN1*) patient HiPSCs	60–90	[121]
Aβ oligomers and Aβ aggregation	Hydrogel-embedded dual SMAD-inhibited differentiation [108]	FAD (*APP* and *PSEN1*) patient fibroblasts	>14	[122]
Aβ plaques Aggregated and abnormal pTau	Component- and environment-controlled differentiation of cerebral organoids	FAD (*PSEN1*) and Down syndrome patient HiPSCs	110	[123]
↑ Tau fragmentation and mislocalization Impaired axonal transport and functionality that can be improved by microtubule stabilization	Matrigel-embedded self-organized differentiation	Familial frontotemporal dementia patient derived HiPSC with *R406W* mutation and isogenic control	60	[124]
Accelerated neuronal differentiation ↑ Synaptic markers ↑ Total tau and pTau	Matrigel-embedded growth factor–directed differentiation of HiPSCs in spinning bioreactor	apoE4^+^ LOAD patient–derived fibroblasts and gene-edited (*apoE4*) healthy control–derived fibroblasts	46	[128]
Early neuronal differentiation Aβ accumulation and elevated pTau	Matrigel-embedded self-organized differentiation	HiPSCs from LOAD patients with *apoE4* mutation	>180	[89]
↑ Secretion of long Aβ peptides (Aβ_40_, Aβ_42_, and Aβ43)	Matrigel-embedded growth factor–directed differentiation of HiPSCs in spinning bioreactor	Fibroblasts from FAD patients with FAD-linked mutations in *APP* or *PSEN1*	100	[125]
Increased Aβ_42_/Aβ_40_ peptide ratios and decreased synaptic protein levels	Matrigel-embedded differentiation in suspension	FAD (*APP* and *PSEN1*) patient HiPSCs	35	[130]

Aβ, amyloid-β; AD, Alzheimer’s disease; apoE4, apolipoprotein E4; APP, amyloid precursor protein; FACS, fluorescence-activated cell sorting; FAD, familial AD; HiPSC, human induced pluripotent stem cell; LOAD, late-onset AD; NPC, neural progenitor cell; PSEN1, presenilin 1; pTau, phosphorylated tau; SMAD, an acronym from the fusion of the *Caenorhabditis elegans* and *Drosophila* genes, *Sma* and *Mad* (mothers against decapentaplegic); 3D, 3-dimensional.

### 2.3. Engineered 3D Models of AD

To overcome limitations associated with organoids for drug discovery, new methods have evolved that use an engineered culturing environment to increase replicability and ease access to phenotypic observations (Table 3). A fixed number of NPCs seeded in concave polydimethylsiloxane (PDMS) microwells with predefined dimensions partially solved this problem (Figure 3a) [139]. Control over cell number and physical confinement yielded neurospheroids with homogenous and reproducible morphology. Toxic effects of Aβ were assessed in this setup in a moderately high-throughput manner. However, long-term morphologic stability in these 3D cultures was difficult to maintain due to their floating nature. In static conditions where there was no culture medium flow through inlet and outlet (Figure 3a), cultures varied in size. With culture medium flow, cultures grew processes and created connections between multiple cultures, defeating the purpose of high-throughput drug screening. The Matrigel embedding stopped this neurite outgrowth [140]. The ReNcell VM with FAD-related mutations in *APP* and *PSEN1* were embedded in Matrigel and cultured in PDMS microwells to yield neurospheroids [140] (Figure 3b). Though at a first glance, these cultures looked like substrate adherent, they were actually pushed down with a top Matrigel layer that nullified the cultures’ buoyancy (Figure 3b). The cytoarchitecture of the cultures was also homogeneous, similar to neurospheroids. These cultures yielded increased deposition of Aβ_42_ and pTau. However, the biggest advantage was in the enhanced level of reproducibility and scalability compatible with the high-throughput screening applications.

In human brain, ECM proteins, such as collagens, laminins, and fibronectins, comprise an optimal mechanical and chemical environment for cells [141]. This environment provides intricate spatiotemporal cues that promote proper cell differentiation and maintenance. Numerous advanced biomaterials [141,142] were developed in order to mimic this environment. Importantly, these fiber-based materials can also be shaped with enhanced precision. In one study, a doughnut-shaped AD model was created from human-induced neural stem cell–differentiated neurons and glia [143] using in silk sponge embedding (Figure 3c) [144,145]. Interestingly, AD induction in this model was performed by infection with herpes simplex virus 1 (HSV-1) which produced multiple AD-related phenotypes, including Aβ plaques, gliosis, inflammation, and impaired functionality. However, as of today, HSV-1 is not a clinically proven causative factor of AD [146]. Nevertheless, this work presents a novel technique to fabricate 3D AD models in vitro with enhanced freedom over cytoarchitecture and morphology.

**Figure 3 brainsci-12-00552-f003:**
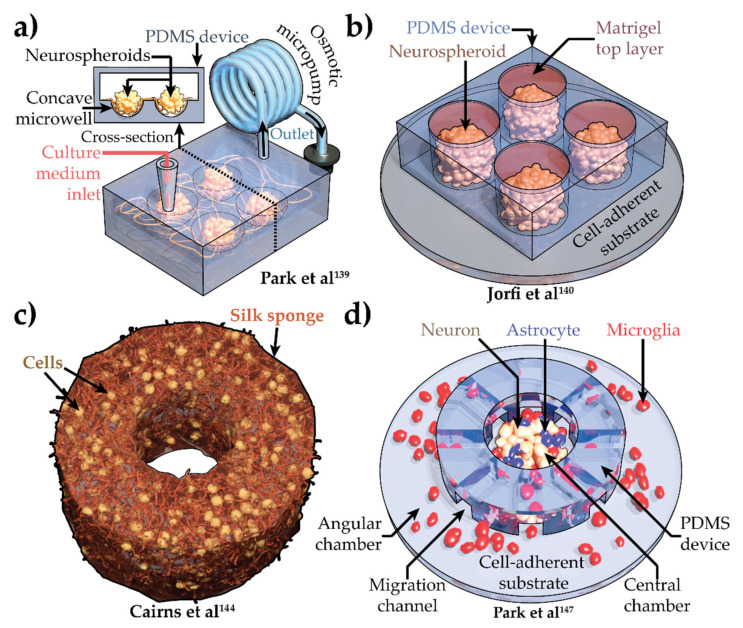
Engineered 3-dimensional (3D) human induced pluripotent stem cell–derived Alzheimer’s disease models. (**a**) 3D culture based microfluidic chip with interstitial flow of culture medium [139]. (**b**) Engineered neurospheroids in PDMS microwell array [140]. (**c**) Silk sponge-embedded 3D neuronal culture [144]. (**d**) 3D Neuron (brown), astrocyte (blue), and microglia (red) tri-culture system in microfluidic device [147]. PDMS, polydimethylsiloxane.

Another important but less-explored aspect in HiPSC-derived 3D AD models is cell-specific roles [68]. Study of AD brain, both in human and mice, clearly indicates the crucial involvement of astrocytes and microglia in AD pathogenesis and progression [66]. HiPSC-based 2D AD models have also proved this relation (Table 1). Cell-type specificity is hard to achieve in systems where all cells are differentiated from NPCs or HiPSCs in a single setup. To circumvent this, the APP with FAD-related mutations *K670N/M671L* (Swedish) and *V717I* (London) was first overexpressed in NPCs [147]. Neurons and astrocytes were then differentiated in Matrigel from these NPCs in the center chamber of the PDMS microfluidic device (Figure 3d). After complete differentiation, immortalized human microglia SV40 cells were seeded in the angular chamber (Figure 3d). These center and angular chambers were connected with channels (Figure 3d), allowing cells to migrate from one chamber to another. Multiple AD phenotypes, including elevated Aβ_40_ and Aβ_42_ secretion, increased tau phosphorylation, and expression of inflammatory cytokines and chemokines, were observed in these cultures. Introduction of microglia in the angular chamber further enhanced neurotoxicity, astrogliosis, and microglia recruitment.

Inflammation is a key player in AD progression [148]. However, only a handful of HiPSC-derived AD models [78,92] captured inflammation in 2D cultures, while organoid models did not exhibit this phenotype at all. Inflammation is a multicellular phenomenon orchestrated by the complicated interplay between neurons, astrocytes, and microglia. Cell type heterogeneity in organoids can be attributed to this deficit. Engineered neuron-astrocyte-microglia triculture system [147] addressed inflammation and microglia-specific mechanisms in AD (Figure 3d). However, the lack of isogeneity across cells in this setup and forced induction of AD can be considered a study limitation.

Combining strengths of these models, a novel yet simple technique was recently developed to create millimeter-sized 3D neuronal cultures from separate populations of HiPSC-derived neurons and astrocytes in PDMS confinement [149] (Figure 4a,b). These cultures are artificial scaffold free, self-assembled, substrate adhered, and reproducible. Though the lack of a scaffold makes the microenvironment less malleable, it also makes these cultures devoid of artificial constructs that might induce unwanted effects. A closer examination of these cultures revealed that astrocytes self-aggregate into a juxtaposed superficial layer, while neurons stay at the core without any prominent necrosis [149]. These astrocytic layers exhibit glial scar-like high expression of glial fibrillary acidic protein, which can be a great tool for comparing anti-inflammatory responses of different treatments. A mathematical model was also developed and experimentally validated that explains this aggregation and provides a tool to accurately forecast cellular aggregation in confinement [150,151]. Because of the system’s substrate adherence, cultures were readily compatible with microelectrode array-based electrophysiologic recording and stable high-resolution optical recording using neuronal optogenetic Ca^2+^ indicator. Both of these recordings revealed developing cortex-like, culture-wide, synchronized neuronal activity or bursts [149]. These bursts were shown to be a useful phenotype to screen for antiepileptic drugs [149]. Compared to conventional 2D and spheroid cultures, the responses of neuronal activities in 3D cultures to multiple antiepileptic drugs better resembled the response observed in vivo. Miniature excitatory postsynaptic current recording via patch clamp indicated that these cultures might have more synapses than conventional 2D cultures resembling an in vivo–like scenario [149]. Moreover, a microfluidic device was integrated into these systems, further assisting with the exploration into axonal trafficking in a high-throughput manner [152] (Figure 4c).

**Table 3 brainsci-12-00552-t003:** Engineered 3D models of AD.

Observed AD Phenotype	Method	Cell Type	AD Source	Experimental Time Point, Days	Reference
Decreased cell viability Synaptic dysfunction	Microwell in enclosed PDMS device	NPC-differentiated neurons	Aβ application	10	[139]
Extracellular Aβ aggregates Elevated intracellular and total pTau	Matrigel (Corning Life Sciences)–scaffolded spheroids in microfabricated microwells	ReNcell VM (ReNeuron Group plc), NPCs	Overexpression of APP variant with FAD mutations in ReNcell VM and FACS	56	[140]
Aβ aggregation, pTau accumulation, increased neuroinflammatory activity, microglial recruitment, axonal cleavage, and inflammatory damage to AD neurons and astrocytes	Matrigel-based 3D culture in engineered PDMS microfluidic device	ReNcell VM–derived neurons, NPC-derived astrocytes, and immortalized human microglia	Overexpression of APP variant with FAD mutations in ReNcell VM and FACS	42	[145]
Amyloid plaquelike formations Gliosis Neuroinflammation Decreased functionality	3D silk sponge ECM [152]	Multiple neuronal and glial subtypes	HSV-1 infection in human NSCs [153]	32	[143]

Aβ, amyloid-β; AD, Alzheimer’s disease; APP, amyloid precursor protein; ECM, extracellular matrix; FACS, fluorescence-activated cell sorting; FAD, familial AD; HSV-1, herpes simplex virus 1; NPC, neural progenitor cell; NSC, neural stem cell; PDMS, polydimethylsiloxane; pTau, phosphorylated tau; 3D, 3-dimensional.

The millimeter-sized systems can further be miniaturized to micrometer range without losing the cortex-like bursts [153] exponentially enhancing this system’s throughput and utility, where synaptic activity is under the investigation. Another version of this system mimicked brain-like clusters and spontaneous neurite alignment in linear cultures [150]. Considering the crucial role of synaptic activity, inflammation, mitochondrial dynamics, and axonal trafficking [56], this system can be highly suitable for modeling AD in vitro. Currently, the development of such systems with cells derived from patients with LOAD are ongoing and promises to shed light into cell- and sex-specific aspects of AD.

**Figure 4 brainsci-12-00552-f004:**
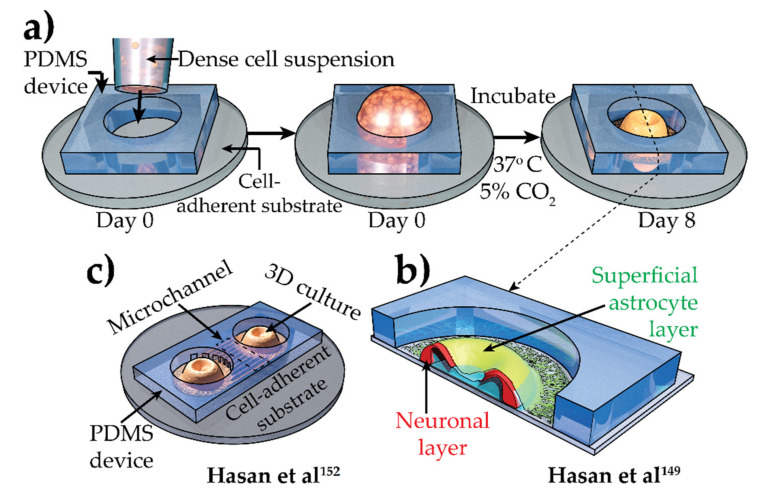
Simplified process to create scaffold-free, confined, adhered 3D culture [149]. (**a**,**b**) Cross-section along dashed line shows juxtaposed neuronal (red) and astrocytic (green) layer. (**c**) Microchannel-connected 3D culture [152]. CO_2_, carbon dioxide; PDMS, polydimethylsiloxane; 3D, 3-dimensional.

Engineered AD model technology is still in its infancy, and patient-derived HiPSCs remain unexplored. However, the initial body of work shows that these models can ease access to different hard-to-observe phenotypes in a reproducible manner. With the recent rise of interest in biofabrication, the development of more sophisticated and advanced engineered 3D cultures can be anticipated in the near future.

### 2.4. HiPSC Xenograft Model

Organoids and engineered 3D cultures aim to mimic brain cytoarchitecture inside the culture. However, the environment surrounding these cultures in a petri dish still remains artificial. In an effort to provide a more natural environment, NPCs were injected into the brains of immunodeficient wild type (WT) mice and mice with FAD related mutations [154]. These NPCs were differentiated from HiPSCs derived from patients with FAD carrying Tau Ex10 + 16 mutation by inhibiting bone morphogenic protein using Noggin [154]. Compared to WT mice, when transplanted in FAD mice, these human neurons exhibited severe neurodegeneration, activated astrocytes, microglial recruitment, hyperphosphorylated tau accumulation, upregulation of genes involved in myelination and downregulation of genes related to memory and cognition, synaptic transmission, and neuron projection. Interestingly, in these human neurons, neurodegeneration occurred before the emergence of tau pathology. The neurodegeneration was not present when instead of the human NPCs, FAD mice were injected with mouse NPCs, indicating the effect of host-transplant incompatibility. In other works, HiPSC derived induced microglia like cells were also transplanted in the brain of FAD mice [155,156]. These microglia like cells migrated towards the Aβ plaques and phagocytosed fibrillar Aβ.

Xenografts provide excellent insight into the in vitro differentiated cell’s behavior in vivo and have opened new avenue in regenerative medicine. However, even with genetically induced immunodeficient mice, current work shows a presence of host-graft incompatibility. This may arise due to mismatched maturation state of cells or due to the specie-specific inherent differences. These issues need to be resolved before HiPSC xenografts could be utilized to comprehensively model AD.

## 3. Future Directions

The multifactorial nature of AD begets the necessity of a commensurate approach. Its unraveled etiology further necessitates the use of HiPSCs and gene editing. Recent advancements in stem cell technology are enhancing the reliability of the generation of different patient-derived cell types. The commensurate effort is also being invested in fabrication methods, which will allow the proper use of these cells. With the established phenotypes of AD, such as altered neuronal network [157], hyperexcitability [59], and inflammation [146], it might be important to induce brain-like organization and structure while modeling AD in vitro. This would require enhanced control over cellular composition and ECM material property. Multiple ECM gels were developed for this purpose. Hydrogel gradient embedded at the bottom of wells of a 96-well plate was used to enhance HiPSC-derived neuronal maturation [158] (Figure 5a). The gradient in hydrogel cross-linking density (low at top and high at bottom) created a soft gel surface and more solid bottom. This allowed cells to gradually get embedded into the scaffold with sequential introduction of glial cells [158]. Multiple other gel-based scaffolds like Matrigel [159,160], polyvinyl alcohol [161], and peptide-based gels [162] are also being explored to create patterned or gradient substrate to promote HiPSC differentiation. However, these technologies have not yet been tested with neuronal cultures.

Additive manufacturing is one of today’s most evolving technologies. This technique, also known as 3D printing, has recently gained a lot of attention due to its controllability and adaptability to different materials. The HiPSC-derived neuronal cultures have also been attempted with this technology [163]. HiPSC-embedded, porous, optimized bioink was poured in a nozzle and extruded in a controlled manner to create a predefined 3D structure (Figure 5b). However, the homogeneity of cells in the final cultures didn’t show a brain-like cytoarchitecture. Since bioink-embedded HiPSCs were differentiated in situ, the introduction of different cell types was also not established. Another technique that stemmed from additive manufacturing [164] is laser-fabricated 3D scaffolds [165] (Figure 5c). Femtosecond lasers [166] are used to polymerize a pattern in a photosensitive resin. Polymerized, hardened scaffold is then extracted and used to hold cells in 3D space. This method is highly controllable, and cultures yielded strong neuronal activity, indicative of healthy synaptogenesis and maturation [165]. However, the complexity of this highly sophisticated and expensive laser system can make the whole process difficult to use.

Along those lines, the electrospun nanofiber-based scaffolds represent a recently developed technology. In this setup, a polymer melt or solution (scaffold material) is loaded in a extrusion nozzle. This nozzle is kept at a high voltage compared to grounded collector. The nozzle is then rapidly moved while extruding nanofibers of scaffold material (Figure 5d) [167,168]. Due to high voltage difference, extruded nanofibers rapidly accelerate towards the grounded collector with needles and create a mesh [164]. The needles allow the proper collection of nanofibers and prevent them from getting squashed on a flat collector. By precise control of the nozzle, nanofibers in the final substrate can be organized either randomly or in a specific pattern. Using this method, poly-ε-caprolactone scaffold (Figure 5d) allowed for successful seeding and differentiation of human NPCs [167]. The induction of neurite alignment was reported in this setup by controlling extrusion nozzle movement [169]. Some of these neurons matured enough to yield action potential. However, this system showed a less favorable environment for neuronal maturation and no brain-like layers. Furthermore, patterned surfaces are being developed, which can provide improved mechanical surface adhesion and replace Matrigel/ECM-embedding methods [170].

**Figure 5 brainsci-12-00552-f005:**
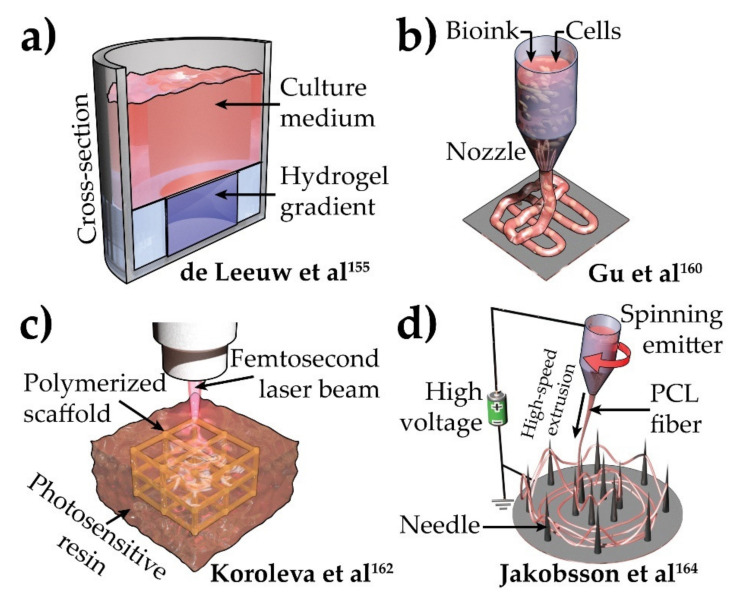
Tentative technologies to be adapted in AD modeling. (**a**) Hydrogel gradient at the bottom of wells of a 96-well plate [155]. (**b**) Three-dimensional (3D) printing [160]. (**c**) Scaffold from femtosecond laser–induced polymerized photosensitive resin [162]. (**d**) Electrospun nanofiber–based scaffold. Cells can be embedded in these scaffolds to create 3D cultures [164]. PCL, poly-ε-caprolactone.

Multiple publications reported the use of Matrigel, hydrogel, microbead [171], and gellan gum–based scaffolding [172,173]. These methods mainly used rodent primary neurons, which are markedly easier to culture than HiPSC-derived cells. Hence, recreation of these setups with HiPSC-derived cells would be crucial before a conclusion can be drawn about their feasibility in modeling AD. Consequently, this review outlined only the work that used HiPSCs or HiPSC-derived cells as building blocks.

## 4. Conclusions

Multiple human-derived in vitro systems are now available to address particular aspects of AD mechanisms. Continual advancement in stem cell technology, bioengineering, and biomaterials further contributes to making these systems increasingly sophisticated and useful. We summarized and discussed strengths and limitations of the currently available systems from a drug discovery perspective. The selection of the model depends on the therapeutic target and the mechanism. While multiple AD phenotypes were successfully replicated in 2D cultures (Table 1), more upstream system-level phenotypes including Aβ plaques were better captured in spheroid systems (Table 2). Engineered 3D cultures further minimize the variability in cellular composition and morphology observed in spheroids and provide enhanced access to cell specific phenotypes (Table 3). However, these engineered systems come with additional complexity and cost. Proper understanding of the strengths and limitations of each system, including the cost, is important before they can be assimilated into a project. The most reasonable and cost-efficient pipeline that addresses therapeutic question may include the progression from a simple 2D human cell cultures to the validation using more complex 3D systems that capture appropriate question-specific subset of human brain cells with the ultimate validation in the appropriate in vivo model to assess full system level attributes for a confident translation into the clinic.

## Figures and Tables

**Figure 1 brainsci-12-00552-f001:**
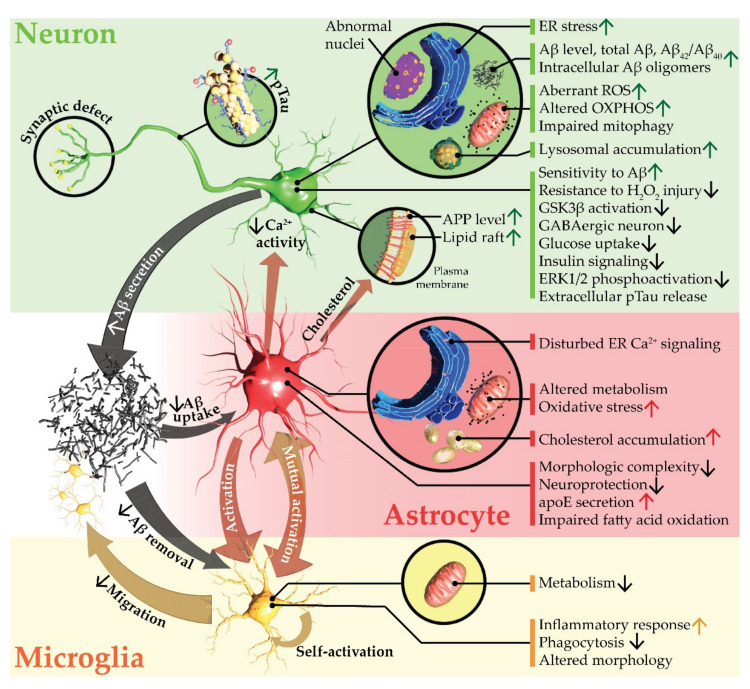
Alzheimer’s disease phenotypes captured in human induced pluripotent stem cell–derived 2-dimensional models. Aβ, amyloid-β; apoE, apolipoprotein E; APP, amyloid precursor protein; Ca^2+^, calcium ion; ER, endoplasmic reticulum; ERK1/2, extracellular signal-regulated kinases 1 and 2; GABAergic, ɣ-aminobutyric acid–mediated; GSK3β, glycogen synthase kinase-3β; H_2_O_2_, hydrogen peroxide; OXPHOS, oxidative phosphorylation; pTau, phosphorylated tau; ROS, reactive oxygen species.

**Table 1 brainsci-12-00552-t001:** HiPSC-derived 2D AD models.

Observed Key AD Phenotype	Differentiation Method	Cell Type	AD Source	Experimental Timeline, Days	Reference
↑ Aβ, pTau levels ↑ GSK3β activation Neuronal endosomal accumulation	Growth factor–guided differentiation of FACS-purified NPCs	Cortical neurons and normal astrocytes	FAD (APP) and LOAD patient fibroblasts	21	[35]
↑ Aβ_42_/Aβ_40_ ratio Intracellular Aβ oligomers ER and cell stress	Small molecule–guided differentiation of EB	Cortical neurons and normal astrocytes	FAD (APP) and LOAD patient fibroblasts	180	[36] ^a^
β-CTF but not Aβ-mediated endosomal abnormality ↓ Endocytosis and transcytosis of APP and lipoproteins	FACS purification of NPCs and neuronal differentiation	Cortical neurons	Gene-edited (PSEN1 ∆E9, APP V717F, or APP SWE) HiPSCs	>21	[45] ^b^
β-CTF but not Aβ-mediated endosomal abnormality	Dual SMAD inhibition and neuronal maturation	Cortical neurons	Multiple FAD-related gene knock-in HiPSCs	80	[40] ^b^
↑ Aβ, pTau levels ↑ GSK3β activation ↑ Sensitivity to Aβ	Differentiated from NPCs, obtained by dual SMAD inhibition of HiPSCs	Cortical neurons and glia	FAD (PSEN1) and LOAD patient fibroblasts	70	[37]
Aberrant cholesterol metabolism–correlated pTau accumulation	Neurons: dual SMAD inhibition and FACS Astrocytes: extended culture of neutrospheres	Neurons and astrocytes	FAD and LOAD patient fibroblasts and gene-edited lines	>35	[44]
↓ Resistance to H_2_O_2_ injury	Serum-free induction of NSCs from HiPSCs and neuronal differentiation	Cortical neurons	HiPSCs from LOAD patients	35	[53]
↑ Aβ and pTau levels GABAergic neuron degeneration	Differentiated from NPCs, obtained by dual SMAD inhibition of HiPSCs	Cortical neurons and glia	Fibroblasts of LOAD patients with apoE4 mutation	>56	[61] ^c^
Lysosomal dysfunction–mediated impaired mitophagy	Dual SMAD inhibition and neuronal maturation	Cortical neurons	FAD patients with PSEN1 A246E mutation–derived fibroblasts	>40	[55]
Several mitochondrial respiratory chain defects Aberrant mitophagy	PSC Neural Induction Medium (Gibco)	NSCs	PSEN1 M146L knock-in HiPSCs	>7	[107] ^d^
Impaired mitophagy	Differentiated from NPCs, obtained by dual SMAD inhibition of HiPSCs [108]	Cortical neurons and glia	HiPSCs from LOAD patients with apoE4 mutation	28	[54]
↑ Aβ_42_/Aβ_40_ ratio ↑ Total Aβ level ↑ Frequency of spontaneous action potentials and evoked activity ↑ Action potential height ↓ Action potential half-width ↓ Neuritic processes length Altered neuronal sodium channel activity ↓ Inhibitory GABA- and PV-positive neurons	Small molecule cocktail	Cortical neurons	CRISPR/CAS9 gene–edited PSEN1 and APP HiPSCs	35	[59]
↑ Vulnerability to glutamate-mediated cell death	Overexpression of transcription factors in NPCs	Cholinergic neurons	LOAD patient fibroblasts	14	[63]
↑ Aβ_42_/Aβ_40_ ratio Altered Ca^2+^ flux	Dual SMAD inhibition with ventralizing agents and maturation in BrainPhys (STEMCELL Technologies Inc.) medium	Cholinergic neurons	FAD with PSEN2 N141I mutation patient–derived HiPSCs	30	[64] ^e^
↑ pTau ↑ ERK1/2 phosphoactivation ↑ Extracellular pTau release	Overexpression of transcription factor in HiPSCs	Cortical neurons	HiPSCs from LOAD patients with apoE4 mutation	38	[43]
Aberrant Aβ or pTau uncorrelated, DNA damage correlated ROS production Altered levels of OXPHOS complexes	Overexpression of transcription factor in HiPSCs	Cortical neurons	LOAD patient fibroblast–derived HiPSCs	21–23	[52]
↑ 4R tau, pTau ↑ Tau aggregation ↑ Neuronal activity ↓ Neurite outgrowth Altered GABAergic gene expression Aberrant differentiation Activation of stress pathways Upregulation of neurodegenerative pathways	Dual SMAD inhibition	Cortical neurons	N279K, P301L, and E10 + 16 mutations in HiPSCs from healthy patients	>70	[62]
↑ Synapse number ↑ Neuronal Aβ_42_ secretion Impaired astrocytic Aβ uptake and cholesterol accumulation Altered microglia morphologies Reduced microglial Aβ phagocytosis	Neurons: overexpression of transcription factor in HiPSCs Astrocytes: differentiated from HiPSC-derived NPCs Microglia: defined serum-free differentiation from HiPSCs	Neurons, astrocytes, and microglia	HiPSCs from LOAD patients with apoE4 mutation	28	[89]
Altered astrocytic mitochondrial metabolism ↑ Oxidative stress Disturbed Ca^2+^ signaling in the astrocytic ER Astrocyte-mediated reduction of neuronal calcium signaling	Differentiated from NPCs, obtained by dual SMAD inhibition of HiPSCs and chemical differentiation	Astrocytes	Early-onset FAD (PSEN1) patient fibroblasts	210	[74]
Impairment in astrocytic fatty acid oxidation	Differentiated from NPCs, obtained by dual SMAD inhibition of HiPSCs and chemical differentiation	Astrocytes	Early-onset FAD (PSEN1) patient fibroblasts	210	[77] ^f^
↓ Morphologic complexity Abnormal localization of key functional astroglial markers Altered nonstimulated release of soluble inflammatory mediators	Chemically defined differentiation method from cortical NPCs	Astrocytes	FAD (PSEN1) and LOAD (apoE4) patient HiPSCs	30	[78]
Less supportive in neuronal survival and synaptogenesis than apoE3 astrocytes	Differentiated from HiPSC-derived NPCs	Neurons and astrocytes	HiPSCs from LOAD patients with apoE4 mutation	45	[79]
↓ Glucose uptake ↓ IGF-1 or insulin responses Altered bioenergetic metabolites and metabolic transcriptomes	Differentiated from HiPSC-derived NPCs	Neurons and astrocytes	LOAD patient fibroblasts and peripheral blood mononucleocytes	60–90	[80]
↑ Inflammatory response ↓ Metabolism ↓ Phagocytosis ↓ Migration	Small molecule–directed differentiation of HiPSCs under defined oxygen conditions	Microglia	FAD (PSEN1 and APP) and LOAD (apoE4) patient HiPSCs	>24	[92]
Mutual activation of microglia and astrocytes	Neurons: small molecule–directed dual SMAD inhibition Astrocytes: lentiviral overexpression of transcriptome factor Microglia: defined chemical differentiation	Neurons, astrocytes, and microglia	FAD (APP) patient HiPSCs	80	[90]
Neuronal synaptic loss, dendrite reduction, axon fragmentation, pTau, Aβ plaque formation, dystrophic neurite around plaque, microglial migration	Aβ oligomer application to triculture with: Neurons: overexpression of transcription factor in HiPSCs Astrocytes: commercially available primary Microglia: defined chemical differentiation	Neurons, astrocytes, and microglia	Neurons: apoE3 or apoE4 Astrocytes and microglia: apoE3	<30	[93] ^g^

Footnotes indicate the therapeutic approach investigated in the study. ^a^ Evaluation of the effect on ER stress or ROS production (DHA, DBM14-26, and NSC23766). ^b^ Application of β-secretase inhibitor rescues endocytosis reduction. ^c^ Validation of a small molecule structure corrector (PH002). ^d^ Autophagy-stimulating drug bexarotene reverts autophagy and mitochondrial abnormality. ^e^ Insulin reverts Aβ42/Aβ40 ratio increase. ^f^ Evaluation of the ameliorating effect of PPARβ/δ-agonist GW0742. ^g^ Seventy small molecule compounds were screened. Multiple hits were found, including DLKi27, indirubin-3′-monoxime, AZD0530, luteolin, curcumin and its derivative J147, demeclocycline HCl, VX-68099, GNE-495100, PF06260933, JNK-IN-8, and anti-Aβ antibodies. Aβ, amyloid-β; AD, Alzheimer’s disease; apoE3, apolipoprotein E3; apoE4, apolipoprotein E4; APP, amyloid precursor protein; Ca^2+^, calcium ion; β-CTF, β-C-terminal fragment; EB, embryoid body; ER, endoplasmic reticulum; ERK1/2, extracellular signal-regulated kinases 1 and 2; FACS, fluorescence-activated cell sorting; FAD, familial AD; GABAergic, ɣ-aminobutyric acid–mediated; GSK3β, glycogen synthase kinase-3β; HiPSC, human induced pluripotent stem cell; H_2_O_2_, hydrogen peroxide; IGF-1, insulin-like growth factor 1; LOAD, late-onset AD; NPC, neural progenitor cell; NSC, neural stem cell; OXPHOS, oxidative phosphorylation; PPARβ/δ, peroxisome proliferator activated receptor-β and -δ; PSEN1, presenilin 1; pTau, phosphorylated tau; PV, parvalbumin; ROS, reactive oxygen species; SMAD, an acronym from the fusion of the *Caenorhabditis elegans* and *Drosophila* genes, *Sma* and *Mad* (mothers against decapentaplegic); 2D, 2-dimensional.

## Data Availability

Not applicable.

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
