# Peer review of "Advances in Recapitulating Alzheimer’s Disease Phenotypes Using Human Induced Pluripotent Stem Cell-Based In Vitro Models"

_brainsci, 2022, doi:10.3390/brainsci12050552_

Round 1

Reviewer 1 Report

This review by Fayad Hasan and Eugenia Trushina highlights the current state of using stem cells-driven 3D models to study Alzheimer's disease, as it offers unique opportunities to fill gaps in existing animal models. In addition, this article provides a scientific perspective on Emerging technology that enables disease models that mimic a brain-like microenvironment based on patient-derived cells used in disease diagnostics and therapeutics. Interestingly, These models retain the individual's unravelled genetic background, yielding clinically relevant disease phenotypes briefly discussed

Author Response

We thank the Reviewer  for positive comments.

Reviewer 2 Report

The review by Hasan et al. has provided a comprehensive account of the various hiPSC models used for deciphering and understanding Alzheimer’s disease phenotypes. It is clearly written and have good descriptive figures and tables summarizing the major points of the review.

General comment:

It would be a great addition to the review in my opinion if the authors can include a separate section on HiPSC transplantation models/ HiPSC xenograft models in mice used for studying Alzheimer’s disease, such as -(Espuny-Camacho, Arranz et al. 2017).

Minor comments:

  1. Alzheimer disease should be spelled as Alzheimer’s disease.
  2. ‘transcriptome factor’ in line 97 should be changed to ‘transcription factor’.
  3. ‘β-C-terminal fragment [β-CTF] fragments’ should be changed to ‘β-C-terminal fragment [β-CTF]’ in line 140, 141.

Reference-

Espuny-Camacho, I., A. M. Arranz, M. Fiers, A. Snellinx, K. Ando, S. Munck, J. Bonnefont, L. Lambot, N. Corthout, L. Omodho, E. Vanden Eynden, E. Radaelli, I. Tesseur, S. Wray, A. Ebneth, J. Hardy, K. Leroy, J. P. Brion, P. Vanderhaeghen and B. De Strooper (2017). "Hallmarks of Alzheimer's Disease in Stem-Cell-Derived Human Neurons Transplanted into Mouse Brain." Neuron 93(5): 1066-1081 e1068.

Author Response

We have added a new section 2.4. HiPSC xenograft model (Line 550-574) according to the Reviewer’s suggestion.

All minor comments were addressed

  1. “Alzheimer disease” is replaced with “Alzheimer’s disease” throughout the manuscript.
  2. ‘transcriptome factor’ in line 97 was changed to ‘transcription factor’.
  3. ‘β-C-terminal fragment [β-CTF] fragments’ was changed to ‘β-C-terminal fragment [β-CTF]’ in line 140, 141.

Reviewer 3 Report

The topic and the figures are so interesting 

Author Response

(The authors gave the same response as above.)

Reviewer 4 Report

This review by Hasan & Trushina aims to summarize existing and developing human iPSC-based models for Alzheimer’s disease. While it is not the only review on this topic recently published, it covers its content in a well-written and thorough way, with clear and accessible explanations.

I found the manuscript to be thoughtfully organized and clearly presented. The writing style is clear, and concepts and terms are defined and explained concisely, making the review accessible to those less familiar with the field. The authors have done a sufficiently deep dive into the literature, covering the current state of human iPSC models at the 2D and 3D levels, key findings, and pros and cons of each approach.

Given that this is a highly active research area, other reviews covering similar territory have been published recently, including this year (e.g. 10.1007/s12015-021-10254-3, 10.4103/1673-5374.335836, 10.3390/biomedicines10020208). However, this manuscript presents the data in a way that builds neatly from 2D models to 3D organoids to engineered 3D systems, with helpful detailed summary tables for each. It stands out as being relevant enough to be of interest to the community despite other reviews in the same space.

My primary recommendation for this review is to improve the clarity and readability of the figures. While I appreciate the effort but into creating detailed visualizations, the figures are overall somewhat visually busy, which detracts from ease of interpretation. Specifically:

Figure 1: This figure is overly busy and crowded with images that are too small to usefully interpret. In particular, the images along the arrows (cholesterol blobs, AB, Ca2+, etc.) are far too tiny to be discerned. The figure would be cleaner and more readable with the labeled arrows alone, and no meaning would be lost.

Figure 2, Panel B: The two edges of the PDMS device facing the reader are confusing. It’s difficult to tell what’s going on between the wells here – are there additional neurospheroids between/below the four wells we can see, or are those just reflections/refractions of those four?

Figure 2, Panel D: This is also a very compact and busy panel. It would help to move the “PDMS device” and “central chamber” labels to the lower right. The viewer then has all the labels pertaining to the device structure at the bottom, and the cell type labels at the top. Thicker arrows, like those in panel C, would help clarify the arrows pointing at the crowded central chamber and the cell types within.

Author Response

We thank the Reviewer  for positive comments. We have implemented all suggestions:

Figure 1: We removed objects around arrows to make the figure easier to read. We also increased font-size and moved astrocyte mediated cholesterol induced increase of APP and lipid raft in the figure to make the concept easy to visualize.

Figure 2, Panel B: Thanks for pointing this out, the refraction property of PDMS device is now removed. Cultures inside wells are now clearly visible.

Figure 2, Panel D: Labels are reorganized and thickness of the arrows is increased.